# The Response to Dietary Leucine in Laying Lens

**DOI:** 10.3390/ani13162659

**Published:** 2023-08-18

**Authors:** Nilva Kazue Sakomura, Matheus Reis, Lucas Pimentel Bonagurio, Bruno Balbino Leme, Gabriel Silva Viana, Mirella Cunha Melaré, Robert Mervyn Gous

**Affiliations:** 1Department of Animal Science, School of Agricultural and Veterinary Sciences, São Paulo State University (UNESP), Jaboticabal, São Paulo 14884-900, Brazil; bonagurio@hotmail.com (L.P.B.); bruno-balbino@hotmail.com (B.B.L.); mirella_melare@hotmail.com (M.C.M.); 2Trouw Nutrition R&D, Trouw Nutrition, El Viso de San Juan, 45215 Toledo, Spain; matheusdpreis@gmail.com; 3Production Systems, Natural Resources Institute Finland (Luke), 31600 Jokioinen, Finland; 4Faculty of Agricultural, Earth and Environmental Sciences, University of KwaZulu-Natal, Pietermaritzburg 3201, South Africa; gous@ukzn.ac.za

**Keywords:** amino acids, dilution technique, efficiency of utilization, egg output, reading model

## Abstract

**Simple Summary:**

The requirement of a laying hen for an essential amino acid depends on its body weight, for maintenance, and on its potential egg mass output. The requirement for maintenance is a greater proportion of the total nutrient requirement in laying hens than in growing animals, and for this reason an accurate prediction of the maintenance requirement of laying hens is essential when partitioning the total daily requirement between the two factors. In this study, which was designed to determine the optimum economic intake of SID-Leucine for laying hens, separate trials were used to measure the requirements for maintenance and for egg mass output. The amount of Leu required for egg production (11.6 mg/g egg) suggests an efficiency of utilization of dietary Leu of 0.79. Because the content of Leu is high in maize, this amino acid is unlikely to be limiting in laying hen feeds in which maize is used as the main cereal.

**Abstract:**

This study aimed to estimate the standardized ileal digestible leucine intake (SID-Leu_i_, mg) in laying hens for maintenance, and to describe the response in laying hens to SID-Leu_i_, thereby providing the information required to determine the optimum economic intake of SID-Leu for laying hens. Two nitrogen balance series, one balanced and the other unbalanced with respect to leucine (Leu), were used to estimate the SID-Leu requirement for maintenance using 36 roosters per series. The roosters were randomly distributed among the six levels of Leu with each level being replicated six times. The six diets were formulated to contain 0.0, 3.5, 6.9, 10.4, 13.9, or 17.4 g/kg of SID-Leu for the unbalanced series and 0.0, 4.0, 8.0, 12.0, 16.0, or 20.0 g/kg of SID-Leu for the balanced series. The SID-Leu_i_ maintenance requirement was calculated as 144 mg/bird d, 66 mg/kg d, 74 mg/kg^0.75^d or 395 mg/BP_m_^0.73^d. For the response trial, 120 individually caged laying hens (63 weeks old) were randomly distributed among eight treatments with 15 replicates. To estimate the SID-Leu_i_ for the population of hens, the Reading Model was fitted to the data using body weight (BW, kg), SID-Leu_i_ and egg output (EO, g). The Reading Model calculated the mg SID-Leu_i_ = 11.6 EOmax + 43.4 BW. The efficiency of SID-Leu utilization for laying hens was estimated to be 79%.

## 1. Introduction

Among the branch-chain amino acids (BCAAs), leucine (Leu) is the most abundant in plant and animal proteins. Depending on its concentration in the diet, Leu may elicit protein synthesis or amino acid degradation. It increases the expression of mTOR protein (mammalian target of rapamycin), characterizing the beginning of the messenger RNA translation into proteins [1]. Conversely, an excess of Leu may stimulate the degradation of valine and isoleucine and affect the metabolism of neutral amino acids [2]. In this sense, adequate but not excessive supplementation of this amino acid is necessary to maximize the efficiency of amino acid utilization.

The requirement of a laying hen for an essential amino acid depends on its genetic potential and may be calculated as the sum of that required for maintenance and for potential egg production. The maintenance requirement is defined as the amount of a given nutrient that allows the conservation of the existing status of the animal, without gain or loss [3], as cited [4]. The requirement for maintenance is a greater proportion of the total nutrient requirement in laying hens than in growing animals [5], and for this reason an accurate prediction of the maintenance requirement of laying hens is essential when partitioning the total daily requirement between the two factors.

The maintenance requirement for amino acids should be measured in adult roosters [6], because in these animals the amino acids are used mainly for maintenance since they are not growing or performing any production activity. Several studies have been conducted to estimate the amino acid requirements for maintenance [7,8,9], but that for Leu has not been studied.

Most studies conducted to estimate amino acid requirements are based on dose response experiments that enable the estimation of the amino acid intake that maximizes egg production [10,11]. Of greater value is a model that enables the response to the amino acid to be described in terms of the requirements for maintenance and egg output, the results of which may be used to determine the optimum economic intake of the amino acid under different economic and biological conditions. The Reading Model [12] was designed to accomplish the above. The model estimates the intake of the test amino acid that meets the requirement of the average individual in the population, based on the mean body weight and mean egg output of the population. On the basis of the marginal cost of the amino acid and the marginal revenue for eggs, the model also estimates the additional amount of the amino acid that is worth feeding above the mean. For this, the variance and co-variance of body weight and egg output in the population needs to be known.

We hypothesize that the Reading Model could be used to predict the optimum economic SID-Leu_i_ (SID-Leu intake) for laying hens. The aim of this study was, therefore, to evaluate the response of laying hens to increasing levels of SID-Leu with the use of the Reading Model and to measure, separately, the maintenance requirement of SID-Leu.

## 2. Materials and Methods

The experiments were conducted at the Laboratory of Poultry Science of the São Paulo State University “Júlio de Mesquita Filho” campus in Jaboticabal. This study was approved (Protocol no: 013783/18) and conducted in accordance with the guidelines of the Animal Ethics Committee of the Animal Science and Veterinary Faculty of São Paulo State University (Brazil).

### 2.1. Experiment I: Leucine Requirement for Maintenance

#### 2.1.1. Birds, Housing and Experimental Design

Two nitrogen balance series were performed to estimate the SID-Leu requirement for maintenance using 36 roosters purchased from a commercial hatchery (Lohmann LSL) per series. The roosters (63 weeks old) were randomly distributed among the six levels of Leu, with each level being replicated six times. Roosters were housed individually in metabolic cages (0.40 × 0.50 × 0.60 m) equipped with a feed trough and nipple drinker. Light program was set at 14 L:10 D h.

#### 2.1.2. Experimental Diets

Three experimental diets were formulated: (1) nitrogen-free isoenergetic (N-free); (2) high-protein and unbalanced feed (summit-1), containing relative deficiency of SID-Leu, used in unbalanced series; and (3) high-protein and balanced feed (summit-2), used in the balanced series to confirm that SID-Leu was the first limiting amino acid in the feed. In the unbalanced series, the summit-1 feed was formulated to exceed by 20% the recommendation of SID-Leu recommended by [13] for roosters, whereas the other amino acids were provided at 50% above the recommendations to obtain a relative deficiency of 30% and maintain the ideal amino acid profile. In the balanced series, an additional dose of L-Leu (nº L8000, Sigma-Aldrich Brasil Ltda, São Paulo, Brazil) was added to the feed, producing a relative SID-Leu deficiency of only 10%. The N-free diet was based on maize starch, sugar and rice husk (Table 1). The amino acid contents of the ingredients used in the formulation were analyzed using near-infrared spectrometry (NIRS) (Evonik Industries AG, São Paulo, Brazil), whereas the total amino acid content in experimental feeds was analyzed using high-performance liquid chromatography (HPLC), and the values obtained were corrected for digestible amino acids using the tabulated coefficients of digestibility [13].

The dilution series was produced by appropriately blending the N-free and high-protein feeds [14]. The levels of SID-Leu for the unbalanced and balanced series were: 0.0, 3.5, 6.9, 10.4, 13.9, and 17.4 g/kg of SID-Leu, and 0.0, 4.0, 8.0, 12.0, 16.0, and 20.0 g/kg of SID-Leu, respectively (Table 2).

#### 2.1.3. Experimental Procedures and Data Sampling

At the start of the trial, roosters were fasted for 48 h to empty the gastrointestinal tract. In this period, they received 60 mL of sucrose diluted in water 1:1 (*v*/*v*), into the crop using a metallic tube, once a day. For the next three days (72 h), each bird was fed 40 g of experimental diet daily by intubation. Body weights were recorded on the morning of the third day. Throughout the three-day experimental period, the N-free diet was provided ad libitum to maintain the energetic homeostasis and minimize metabolic and endogenous losses of roosters. Feed intake was calculated over the 72-h period as the sum of feed provided by intubation and the amount of N-free diet consumed. The intake of L-Leu was calculated in terms of mg/kg body weight d.

As a marker for excreta, 1% iron oxide was used at the beginning and end of the excreta collection period. Trays covered with plastic were placed under the metabolic cages to collect the excreta which were collected twice a day, weighed (at 8 and 16 h), and placed in plastic containers kept at −20 °C.

At the end of the assay period, three roosters were randomly chosen to measure body composition. Roosters to be sampled were fasted for eight hours to empty the gastrointestinal tract, whereafter they were sacrificed, weighed, plucked, and stored at −20 °C for further analysis.

#### 2.1.4. Laboratory Analysis

The amount of excreta from each rooster was weighed after three days of collection. The excreta collected were homogenized in plastic bags for 3 min, and subsequently were sampled and weighed in Petri dishes. The frozen carcass was cut (Delta Grill DG-1003, São Paulo, Brazil) and ground using a meat grinder (CAF 22 DSM, São Paulo, Brazil). The ground carcass was homogenized in plastic buckets, after which samples were taken and stored in an Ultra freezer (ColdLab Ultra Freezer CL374-86V, Piracicaba, São Paulo, Brazil) for 24 h at −80 °C, then lyophilized (Edwards Supermudulyo-220 freeze dryer, Thermo Electron Corp., Waltham, MA, USA) at a temperature of −80 °C and 10^−1^ atm pressure. Carcasses were also homogenized. Milled samples were lyophilized (−80 °C; −80 kPa; VLP20, Thermo Fisher Scientific, Inc., Waltham, MA, USA), for 72 h. Subsequently, samples were ground in a micro mill (A11 BASIC—IKA, Sao Paulo, Brazil), for 2 min. Dried samples of the experimental diets, excreta, and whole carcass (free of feathers) were analyzed for dry matter, crude protein by Kjeldahl (method 2001.11), and ether extract (method 920.39), according to AOAC (1995) procedures.

#### 2.1.5. Statistical Analysis

Nitrogen balance (NB) was calculated as the difference between intake and excreted N and expressed as a function of SID-Leu_i_. The *x*-axis intercept was assumed to be the maintenance requirement, expressed using four scales, or units: (1) mg SID-Leu_i_/bird, (2) mg SID-Leu_i_/kg body weight (BW), (3) mg SID-Leu_i_/kg metabolic weight (BW^0.75^); and (4) mg SID-Leu_i_/kg metabolic body protein weight at maturity (BP_m_^0.73^) [7]. Data were considered statistically significant when *p* < 0.05. The simple linear regression between NB and SID-Leu_i_ was performed using the PROC REG from SAS (SAS Institute Inc., Cary, NC, USA).

To test if the intercepts of the responses to balanced and unbalanced series differed, PROC REG of software SAS (SAS Institute Inc., Cary, NC, USA) was applied. Where the intercepts did not differ, the maintenance requirement could be regarded as being the same, and data from both assays could be combined to increase the number of observations. A suitable regression was again performed using data from both series to estimate the final maintenance value. For each adopted scale, a second-degree polynomial regression (QP) best fitted the data.

### 2.2. Experiment II: Response of Laying Hens to Digestible Leucine

#### 2.2.1. Birds, Housing and Experimental Design

One hundred and twenty Lohmann LSL laying hens, 63 weeks old at the start of the trial, were used to measure the response of laying hens to digestible leucine. Seven levels of SID-Leu were used in the dilution series, and an additional treatment was applied in which the feed with the lowest content of Leu was supplemented with synthetic Leu to test whether Leu was the first-limiting amino acid in the series [14]. The eight treatments were randomly distributed among the hens, resulting in 15 replicates per treatment. A single laying hen was the experimental unit.

Laying hens were placed individually in stainless-steel cages (50 × 45 × 40 cm) each equipped with a trough feeder and nipple drinker. They received water and feed *ad libitum* throughout the experimental period, and the lighting program was set at 16 L:8 D. Control of temperature, humidity and air renewal were carried out automatically using pad cooling and exhaust fans, being programmed according to breeder recommendations.

#### 2.2.2. Experimental Diets

Two iso-energetic basal feeds (high-protein and N-free) were formulated (Table 3), and these were appropriately blended to produce the eight experimental feeds [14] used in the trial. The high-protein feed contained 1.20 times the SID-Leu levels suggested by [13] and 1.40 times the requirement for the other amino acids, to obtain a relative Leu deficiency of 20% and maintain a fixed amino acid profile throughout the series. Seven feeds with increasing levels of SID-Leu represented the dilution series (6.0, 7.0, 8.0, 9.0, 10.0, 11.0, 12.0 g/kg, Table 4), whilst the eighth treatment (7.0 g/kg of SID-Leu) was made by adding an additional dose of L-Leu to the feed with the lowest level of Leu in the dilution series.

The high-protein feed was formulated based on digestible amino acid values according to the digestibility coefficient from AMINODAT^®^ [15]. Feed ingredients were analyzed for total amino acid content using NIRs (Evonik Industries AG, São Paulo, Brazil), and the digestible amino acid contents were calculated according to the digestibility coefficient from AMINODAT^®^ [15]. After blending, each feed was sampled to analyze amino acid content, using high-performance liquid chromatography (HPLC).

#### 2.2.3. Data Collection

The study was conducted over a period of 10 weeks, with 6 weeks being used for adaptation and the last 4 weeks for data collection. Laying hens were fed at the same time each morning, and the physical form of diet was mash. At the end of each week, the leftovers were weighed to quantify the weekly consumption of feed. The body weight (BW, kg) of hens was measured on the first, sixth, and tenth weeks of the assay. Egg production was recorded daily, and egg weight was measured on three days each week. The variables collected were egg production (EP, %), feed intake (FI, mg/bird d), and egg weight (EW, g). Egg yolk and albumen weights were measured fortnightly. Egg output (EO) was calculated by multiplying EP by EW.

#### 2.2.4. Statistical Analysis

Dunnett’s one-tailed *t*-test was used to compare the response of laying hens consuming the balanced feed (feed 8) against the treatment with the lowest level of SID-Leu, using PROC GLM option DUNNETTL, respectively. A significance of 5% was accepted as statistically different. Amino acid intake, EO, and BW were analyzed as one-way ANOVA by SAS^®^ System version 9.3 (SAS Institute Inc., Cary, NC, USA). The Reading Model was used to apportion the amino acid intake (mg/bird/d) to EO and BW, according to procedures available in the EFG software [16]. The Reading Model equation used for an individual hen was as follows:
(1)$$\mathrm{Iaa}=a\bar{E}O_{\max}+b\bar{\mathrm{BW}}$$
where Iaa is the estimated intake of SID-Leu (mg/bird d); *a* is the amount of amino acid required per g egg output (mg/g); *b* is the amount of amino acid required per unit of body weight (mg/kg); ${\bar{\mathrm{EO}}}_{\max}$ is the mean maximum egg output estimated (g/d); $\bar{\mathrm{BW}}$ is the mean *BW* (kg). Assuming that both *EO* and *BW* are normally distributed, the observed variance (σ^2^) of *BW* and *EO* is included in the EFG software [16]; thus, considering the basic property of variance, which states that if a variable is scaled by a constant, the variance is multiplied by the square of the same constant, the final result, for a population of hens, is the expression:
(2)$$Iaa=a\bar{E}O_{\max}+b\bar{BW}+Z\sqrt{a^{2}\sigma^{2}{\bar{EO}}_{\max}+b^{2}\sigma^{2}\bar{BW}}$$
where *Z* represents the extra amino acid necessary to meet the requirement of 1 standard deviation of the population above the requirement of the average individual in the flock [17]. The value is calculated with MS Excel^®^ function “NORM.S.INV”, which converts the proportion of the population to standard deviation value (0 = 50%; 1 = 84%; and 2 = 97.7%).

## 3. Results

### 3.1. Maintenance Requirement for Leu

The results of the N-balance trial are presented in Table 5, Table 6, Table 7 and Table 8. In each Table, BW, SID-Leu_i_ intake, N intake (NI), N excretion (NE) and NB of roosters are presented, but using different units (mg/bird, mg/kg, mg/kg^0.75^ and mg/BP_m_^0.73^) in each Table. Although the N-free diet was available to each bird ad libitum, it was the amount of the test feed, fed by intubation, that was used to calculate the amount of L-Leu consumed each day. In general, NB became positive above a dietary Leu content of 4.0 g/kg.

The intercepts of the unbalanced and balanced series did not differ from each other (*p* > 0.05). Thus, it was evident that the response in NB was primarily to SID-Leu_i_ and not to the protein content of the feeds. Consequently, data from both assays were combined and this resulted in a polynomial equation for each adopted scale. The SID-Leu maintenance requirement, using different scales, is presented in Table 9. Figure 1 illustrates the NB considering data from both assays, expressed in mg/BP_m_^0.73^.

### 3.2. Response in Egg Production to Leu Intake

Laying hens fed the balanced feed (feed 8) showed an improvement in EO compared to laying hens given feed 7 (*p* < 0.05), confirming that SID-Leu was the first limiting nutrient in the feed (Table 10). The mean initial body weight of laying hens was 1.6 kg and, at the end of the experiment, those hens fed 6 g SID-Leu/kg had suffered a 5% weight loss. The results in Table 10 show an increase in BW, ROL, EW and, hence, EO, as SID-Leu_i_ was increased.

### 3.3. Leucine Requirement Considering the Variation of Population

The Reading Model was fitted using mean Leu_i_, BW and EO for each treatment and standard deviations: *EO*_max_ = 68 g/hen d, σEOmax = 5.5 g/hen d, BW = 1.61 kg and σBW = 0.20 kg. The fitted coefficients of response for EOmax and BW were 11.6 mg/g EO and 43.4 g/kg B*W.*

The Leu content of an egg is reported to be 9.17 mg/g [18]; thus, the efficiency of utilization of SID-Leu for egg production is 0.79. To meet the requirement of the average individual in the population, where Z = 0, the SID-Leu_i_ would be 858 mg/bird d.

## 4. Discussion

The amino acid intake required for maintenance may be expressed in different units, e.g., mg/bird, mg/kg BW, mg/kg BW^0.75^, or mg/kg BPm^0.73^. Expressing this as mg/bird ignores the weight (or size) and the physiological state of the bird, whereas units in mg/kg and mg/kg BW^0.75^ correct this issue. Although frequently used, the last two do not consider the body composition of the bird, attributing to the amino acids the function of maintaining the reserves of body lipid [7]. As mentioned by Emmans [6] and evidenced by Bonato et al. [9], the amino acid requirement for maintenance is more accurately described when expressed as a proportion of body protein content rather than body weight or metabolic body weight, since body lipid is assumed not to require amino acids for maintenance purposes.

The above scales were used to express the maintenance requirements for SID-Leu estimated in this study in which roosters were used as the animal model. The requirements were estimated to be 144 mg/bird, 66 mg/kg, 74 mg/kg BW^0.75^ and 395 mg/kg BPm^0.73^. These can be compared with reports in the literature, such as those by Leveille and Fisher [19], who suggested a requirement for SID-Leu of 124 mg/kg, Dorigam et al. [20], who reported 299 mg/bird, 50 mg/kg BW, 78 mg/ kg BW^0.75^, and 282 mg/ kg BPm^0.73^, and by Sakomura et al. [5], who reported the maintenance requirement of SID-Leu for broiler breeders as being 82 mg/d or 51 mg/kg BW^0.75^ d. The Reading Model used in the present trial calculated the maintenance requirement to be 43.4 mg/kg BW, which is not vastly different from the value 66 mg/kg measured using the N-balance method applied here.

One of the drawbacks of the Reading Model is that maintenance is expressed as a proportion of body weight and not of mature body protein weight, but without knowledge of the protein content of the hens being used in the trial it would not be possible to apportion the amino acid consumed between egg output and body protein content. Hence, the maintenance requirement was assessed using adult roosters, eliminating the need to consider egg production, and being able to evaluate the body protein content of the birds used in the trial because far fewer birds are used in such an evaluation.

An advantage of the dilution technique used here is the ability to verify that the test amino acid is always the first-limiting nutrient in the series [21,22]. This test was applied in both the N-balance assay and in the Leu response trial with laying hens, the results of these tests demonstrating that SID-Leu was the limiting nutrient in both instances. Previous studies also adopted this technique to validate their findings [8,9].

The maximum egg production response measured in this trial was impressive (Table 10), with rates of lay above 90%, egg weights above 62 g and maximum egg mass output close to 62 g/d. Performance declined substantially as the Leu content of the feed was reduced, although feed intake was not affected until the lowest level of Leu was fed, when intake decreased. The resulting response in rate of laying, egg weight and egg output enabled the Reading Model to be accurately fitted to the data.

The optimum economic amount of an amino acid to be fed to a population of hens depends on the marginal cost of the amino acid and the marginal revenue for eggs, these economic factors varying in different countries and at different times of the year. It is possible to determine this optimum intake by making use of the Reading Model [12], which calculates the additional amount of the amino acid that is worth feeding above the mean for the population. For example, to meet the SID-Leu requirement of hens up to one SD above the mean (84% of the population), it would be necessary to add an additional 64.3 mg/bird d to that required by the average individual, i.e., 922 mg/bird d. Two SDs above the mean, representing 97.7% of the population, would require an additional 129 mg, or 987 mg/bird d of SID-Leu. As an exercise, the results of Leu for maintenance from Trial 1 were used in the Reading Model equation instead of the coefficient determined by fitting the model. Thus, applying the same standard errors used above, the SID-Leu requirements were calculated, based on the different scales and percentages of population (Table 11). To obtain the requirements in mg/BPm^0.73^, the body protein at maturity (BPm) used here was that published by Alves et al. [23].

Currently, the marginal revenue for eggs in Brazil varies depending on whether the eggs are sold wholesale (0.0015 c/egg) or retail (0.0052 c/egg) and this would impact on the optimum economic intake of an amino acid. The marginal cost of an amino acid generally increases due to the additional cost involved in supplying increasing amounts of the amino acid in the feed. In the case of Leu, however, the marginal cost declines with an increase in Leu content in the feed. When using the feed ingredients typically included in layer feeds in Brazil (maize and soybean oilcake), a feasible formulation is not possible at the level of inclusion required by a hen consuming 110 g feed/d to meet the requirement for the average individual in the population (858/110 = 7.8 g Leu/kg). The lowest feasible level of Leu in the feed under these conditions is 10.6 g/kg, and the highest, 12.5 g/kg. The Leu content required by the average hen in the population, of 7.8 g/kg, is thus well below the minimum feasible content. Because the cost of feed declines with increasing Leu content, from an inclusion rate of 10.6 to 12.5 g/kg, above which the solution is again infeasible, the optimum economic intake of Leu in Brazil currently, for a hen consuming 110 g/d, is 1375 mg/d. Unlike the case with all other essential amino acids, the Leu content of the main ingredient used in laying hen feeds in some countries, namely, maize, is so high that inclusion rates are well above the amount required biologically. Where maize is the main cereal source used in laying hen feeds, the optimum economic intake of Leu is, therefore, not calculated using the method applied to other essential amino acids but is dictated by the least-cost formulation system.

The use of the Reading Model to describe the response of a population of hens to a limiting amino acid has theoretical support. Taking account of the variation in body weight and maximum potential egg output in the population, as well as the marginal cost of the amino acid and marginal revenue for eggs, makes it possible to determine the optimum economic amino acid intake for flocks varying in body weight and potential egg output. To implement the results of these calculations in practice it is necessary to be able to predict the amount of the given feed that the birds will consume. In the case of Leu, the prediction of feed intake is not necessary, as the amount to be incorporated into the feed is dictated by the feasible amount that can be included in the least cost formulation.

## 5. Conclusions

The maintenance requirement for SID-Leu, using three different scales, was estimated to be 66 mg/kg, 74 mg/kg^0.75^, and 395 mg/BPm^0.73^, respectively. The amount of Leu required for egg production (11.6 mg/g egg) suggests an efficiency of utilization of dietary Leu of 79%. Assuming an average population with 1.6 kg in body weight, 60 g in egg mass, and 100 g in feed intake, a minimum of 0.81% of dietary SID-Leu (0.11% for maintenance plus 0.70% for egg mass) is necessary to meet the requirement of 50% of the population. Because the content of Leu is high in maize, this amino acid is unlikely to be limiting in laying hen feeds in which maize is used as the main cereal.

## Figures and Tables

**Figure 1 animals-13-02659-f001:**
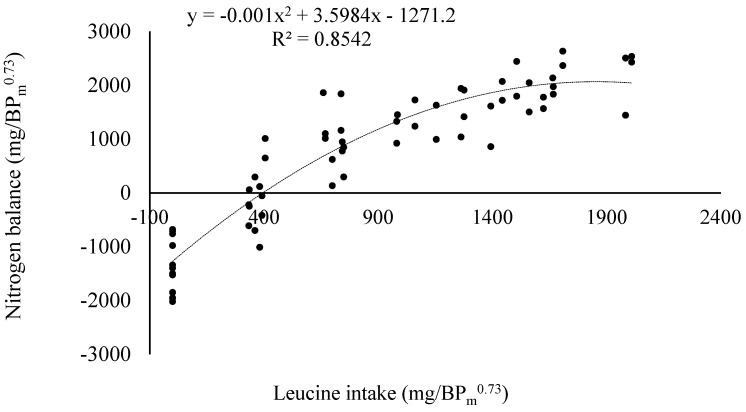
Graphical representation of a polynomial equation between nitrogen balance and SID-Leucine intake considering the protein content of the body of birds.

**Table 1 animals-13-02659-t001:** Composition of basal diets used in the N-balance trial with cockerels.

Ingredients	Summit-1 (g/kg)	Summit-2 (g/kg)	N-Free (g/kg)
Corn (78.8 g of CP ^2^/kg)	184	184	0
Maize starch	0	0	499
Soybean meal (450 g of CP/kg)	227	227	0
Sugar	0	0	370
Peanut meal (482 g of CP/kg)	90	90	0
Soy protein isolate (627 g of CP/kg)	72.8	72.8	0
Maize gluten meal (600 g of CP/kg)	16.4	16.4	0
Potassium carbonate	0	0	14.2
Rice husk	0	0	70.8
Rice broken	250	250	0
Soybean oil	65.4	65.4	0
Dicalcium phosphate	21.2	21.2	27
Limestone	9.2	9.2	7.2
Salt	3.3	3.3	8.2
Sodium bicarbonate	7	7	0
Premix mineral and vitamin ^1^	2	2	2
DL-Methionine (99%) ^3^	11.4	11.4	0
L-Lysine (78%)	11.1	11.1	0
L-Threonine (98%)	8.6	8.6	0
L-Tryptophan (99%)	2.5	2.5	0
L-Valine (98%)	6.4	6.4	0
L-Arginine (98.5%)	1.9	1.9	0
L-Leucine (98%)	0	2.8	0
L-Isoleucine (99%)	6.5	6.5	0
Choline chloride (60%)	1	1	1
Salinomycin	0	0	0.5
Inert filler	2.8	0	0
Total	1000	1000	1000
Composition (g/kg)			
Crude protein	275	277	2.24
Metabolizable energy (MJ/kg)	13.4	13.5	13.4
SID-Lysine	19.1	19.1	0
SID-Methionine	14.1	14.1	0
SID-Met. + Cys.	17.4	17.4	0
SID-Threonine	15.6	15.7	0
SID-Tryptophan	5.05	5.05	0
SID-Arginine	18.8	18.8	0
SID-Isoleucine	15.5	15.5	0
SID-Leucine	17.4	20.0	0
SID-Histidine	5.39	5.39	0
SID-Phe. + Tyr.	19.2	19.2	0
SID-Valine	16.0	16.0	0

^1^ Content per kg of feed—vit. A = 10,575 IU; Vit. D3 = 2554 IU; Vit. K = 1.8 mg; Vit. E = 14.87 mg; Vit. B1 = 2.00 mg; Vit. B2 = 4.5 mg; Vit. B6 = 2.50 mg; Vit. B12 = 2.00 mg; Niacin = 30.00 mg; Folic acid = 0.75 mg; Calcium pantothenate = 11.74 mg; Biotin = 0.01; Iron = 43.44 mg; Zinc = 43.35 mg; Copper = 8.56 mg; Manganese = 56.00 mg; Iodo = 0.56 mg; Selenium = 0.34 mg; Butylated hydroxytoluene = 4.20 mg; Salinomycin (120 g of salinomycin/kg of product) = 550 mg. ^2^ CP is crude protein. ^3^ Values inside the parenthesis for each crystalline amino acid are the purity reported by the producer.

**Table 2 animals-13-02659-t002:** Blending of the summit and N-free diets to produce the unbalanced and balanced series of feeds used in the N-balance trial with cockerels.

	Unbalanced Series			Balanced Series		
Treatment	Summit (g)	N-Free (g)	Leu ¹ (g/kg)	Summit (g)	N-Free (g)	Leu (g/kg)
1	0	40	0.0	0	40	0.0
2	8	32	3.5	8	32	4.0
3	16	24	6.9	16	24	8.0
4	24	16	10.4	24	16	12.0
5	32	8	13.9	32	8	16.0
6	40	0	17.4	40	0	20.0

¹ SID-Leucine.

**Table 3 animals-13-02659-t003:** Composition and nutritional content (g/kg) of experimental diets used in the Leu response trial with laying hens.

Ingredients	Summit	Nitrogen-Free
Corn (78.8 g of CP ^1^/kg)	410	
Wheat bran	146	
Soybean meal (540 g of CP/kg)	216	
Maize gluten meal (600 g of CP/kg)	4.4	
Sugar		100
Limestone	105	105
Maize starch		709
Soybean oil	65.9	
Dicalcium phosphate	13.3	20.4
Rice broken		47.0
Salt	4.7	3.8
L-Lysine HCl (78%) ^2^	5.9	
DL-Methionine (99%)	8.1	
L-Threonine (98.5%)	4.8	
L-Valine (98%)	5.6	
L-Tryptophan (98%)	1.2	
L-Arginine (98%)	3.2	
L-Isoleucine (99%)	3.3	
Premix mineral and vitamin ^3^	2.0	2.0
Potassium carbonate		9.5
Potassium chloride		
Choline chloride (60%)	1.0	1.0
BHT		
Total	1000	1000
Calculated composition (g/kg) ^4^		
Metabolizable Energy (MJ/kg)	12.13	12.13
Crude Protein, g/kg	182 (181) *	1.5
Calcium	44.0	45.0
Available phosphorus	3.7	3.7
Sodium	2.2	2.1
Potassium	7.3	5.3
SID-Met. + Cys.	12.4 (10.9) **	0.0
SID-Methionine	10.6 (8.2) **	0.0
SID-Lysine	11.5 (10.9) **	0.0
SID-Threonine	9.7 (9.5) **	0.0
SID-Tryptophan	2.9	0.0
SID-Isoleucine	9.0 (8.1) **	0.0
SID-Valine	11.7 (11.5) **	0.0
SID-Leucine	12.0 (14.1) **	0.0
SID-Arginine	12.6 (11.9) **	0.0

^1^ CP is crude protein. ^2^ Values inside the parenthesis for each crystalline amino acid are the purity reported by the producer. ^3^ Content per kg of feed—Folic acid = 2400 mg; Pantothenic acid = 30 g; Biotin = 160 mg; Butylated hydroxytoluene = 100 mg; Niacin = 84 g; Selenium = 600 mg; Vit A = 200,000.000 UI; Vit B1 = 5000 mg; Vit B12 = 36,000 mcg; Vit B2 = 1300 g; Vit B6 = 7000 mg; Vit D3 = 5000.000 UI; Vit E = 375,000 UI; Vit K3 = 4000 mg; Manganese = 150 g; Zinc = 140 g; Iron = 90 g; Copper = 15 g; Iodine = 15 mg. ^4^ Calculated composition of energy, crude protein and amino acid content. * Values between parentheses refer to analyzed amino acid. ** Calculated value based on the analyzed amino acid content (ileal digestibility coefficient for amino acids presented in AMINODAT^®^ [15]).

**Table 4 animals-13-02659-t004:** Blending proportions of summit and N-free diets and the calculated concentration of SID-Leucine.

Treatment	Summit	N-Free	Leucine (g/kg) ^1^
1	0.50	0.50	6.0 (7.8) ^1^
2	0.58	0.42	7.0 (9.6)
3	0.67	0.33	8.0 (9.7)
4	0.75	0.25	9.0 (11.5)
5	0.83	0.17	10.0 (13.5)
6	0.92	0.08	11.0 (14.1)
7	100	0.00	12.0 (14.8)
8 ^2^	0.50	0.50	7.0 (9.1)

^1^ Values in parenthesis refer to analyzed amino acid; ^2^ T8 = T1 + 1.00 g L-Leucine/kg of feed.

**Table 5 animals-13-02659-t005:** Results of N-balance trial based on the scale mg/bird/d ^1^.

Leu Content (g/kg)	BW (kg) ^2^	Leu Intake (mg/Bird/d) ^3^	NI (mg/Bird) ^4^	NE (mg/Bird) ^5^	NB (mg/Bird/d) ^6^
Unbalanced sries					
0.0	1.95	0.00	0.00	1519	−506
3.5	1.93	121	1025	1280	−85
7.0	1.92	242	2050	703	449
10.4	1.99	363	3075	1371	397
13.9	1.82	485	4099	1795	355
17.4	2.05	606	5124	2306	654
SEM ^7^	0.05	34.8	295	203	73.6
Balanced series					
0.0	1.83	0.00	0.00	1043	−347
4.0	1.94	139	1039	1003	11.9
8.0	2.15	278	2078	1152	308
12.0	1.83	417	3118	1611	502
16.0	1.90	556	4157	2169	662
20.0	1.89	695	5197	2872	775
SEM	0.04	39.0	292	210	75.5

^1^ Each value is the mean of 6 replicates. ^2^ Body weight; ^3^ SID-Leucine intake; ^4^ Nitrogen intake; ^5^ Nitrogen excretion; ^6^ Nitrogen balance; ^7^ Standard error of the mean.

**Table 6 animals-13-02659-t006:** Results of N-balance trial based on the scale mg/kg/d ^1^.

Leu Content (g/kg)	BW (kg) ^2^	Leu (mg/kg/d) ^3^	NI (mg/kg/d) ^4^	NE (mg/kg/d) ^5^	NB (mg/kg/d) ^6^
Unbalanced series					
0.0	1.95	0.00	0.00	789	−263
3.5	1.93	62.8	532	709	59.1
7.0	1.92	126	1072	574	230
10.4	1.99	183	1551	833	285
13.9	1.82	266	2249	973	193
17.4	2.05	296	2508	1120	321
SEM ^7^	0.05	3.56	30.1	101	48.2
Balanced series					
0.0	1.83	0.00	0.00	567	−189
4.0	1.94	71.7	536	508	9.50
8.0	2.15	129	969	533	145
12.0	1.83	229	1712	882	276
16.0	1.90	294	2201	1155	348
20.0	1.89	367	2746	1516	410
SEM	0.04	3.21	23.8	110	38.5

^1^ Each value is the mean of 6 replicates. ^2^ Body weight; ^3^ SID-Leucine intake; ^4^ Nitrogen intake; ^5^ Nitrogen excretion; ^6^ Nitrogen balance; ^7^ Standard error of the mean.

**Table 7 animals-13-02659-t007:** Results of N-balance trial based on the scale mg/kg^0.75^/d ^1^.

Leu Content (g/kg)	BW (kg) ^2^	Leu (mg/kg^0.75^/d) ^3^	NI (mg/kg^0.75^) ^4^	NE (mg/kg^0.75^) ^5^	NB (mg/kg^0.75^/d) ^6^
Unbalanced series					
0.0	1.65	0.00	0.00	853	−237
3.5	1.64	74.0	626	776	36.3
7.0	1.63	149	1260	668	271
10.4	1.67	217	1840	987	339
13.9	1.57	309	2613	1361	225
17.4	1.71	354	2998	1610	383
SEM ^7^	0.03	3.21	27.1	106	52.1
Balanced series					
0.0	1.58	0.00	0.00	792	−220
4.0	1.64	84.6	633	723	10.2
8.0	1.77	156	1172	775	175
12.0	1.57	265	1988	1025	321
16.0	1.61	345	2580	1352	409
20.0	1.61	430	3221	1779	480
SEM	0.02	2.81	21.0	128.0	45.4

^1^ Each value is the mean of 6 replicates. ^2^ Body weight; ^3^ SID-Leucine intake; ^4^ Nitrogen intake; ^5^ Nitrogen excretion; ^6^ Nitrogen balance; ^7^ Standard error of the mean.

**Table 8 animals-13-02659-t008:** Results of N-balance trial based on the scale mg/BPm^0.73^/d ^1^.

Leu Content (g/kg)	BW (kg) ^2^	Leu Intake (mg/kg^0.73^/d) ^3^	NI (mg/kg^0.73^) ^4^	NE (mg/kg^0.73^) ^5^	NB (mg/BPm^0.73^/d) ^6^
Unbalanced series					
0.0	0.355	0.00	0.00	4309	−1436
3.5	0.353	343	2905	3623	−239
7.0	0.352	690	5842	2063	1259
10.4	0.361	1009	8538	3819	1110
13.9	0.339	1431	12,106	5255	1042
17.4	0.369	1645	13,918	6231	1779
SEM ^7^	0.01	14.9	126	493	195
Balanced series					
0.0	0.340	0.00	0.00	3059	−1019
4.0	0.354	392	2936	2795	47.0
8.0	0.382	728	5449	3003	815
12.0	0.339	1231	9208	4747	1486
16.0	0.348	1599	11,958	6266	1897
20.0	0.348	1996	14,930	8246	2228
SEM	0.005	12.7	95.1	593	211

^1^ Each value is the mean of 6 replicates. ^2^ Body weight; ^3^ SID-Leucine intake; ^4^ Nitrogen intake; ^5^ Nitrogen excretion; ^6^ Nitrogen balance; ^7^ Standard error of the mean.

**Table 9 animals-13-02659-t009:** Maintenance requirement for SID-Leucine estimated from polynomial regression.

Scales	Equation	R^2 (1)^	Maintenance
mg/bird d	y = −0.0024x^2^ + 3.3315x − 429.03	0.79	144
mg/kg d	y = −0.0058x^2^ + 3.6522x − 214.60	0.78	66
mg/kg^0.75^ d	y = −0.0044x^2^ + 3.4346x − 230.49	0.80	74
mg/BP_m_^0.73^ d	y = −0.001x^2^ + 3.5984x − 1271.2	0.85	395

^(1)^ Determination coefficient.

**Table 10 animals-13-02659-t010:** Mean (±standard error of mean) responses in feed intake (FI), SID-Leu intake, rate of lay (ROL), egg weight (EW), egg output (EO) and final body weight (BW) of laying hens fed diets containing increasing levels of SID-Leu.

Leu Content (g/kg)	FI (g/d)	SID-Leu (mg/d)	ROL (%)	EW (g)	EO (g/hen/d)	BW (g)
6.0	96.6 ± 12	579 ± 69.8	75.0 ± 5.8	59.4 ± 3.7 *	44.4 ± 3.2 *	1528 ± 191 *
7.0	102 ± 7.3	710 ± 50.8	85.7 ± 11.4	62.6 ± 3.5	54.8 ± 6.7	1451 ± 405
8.0	102 ± 9.2	815 ± 73.6	89.3 ± 9.1	62.8 ± 2.0	58.3 ± 5.3	1502 ± 419
9.0	102 ± 7.3	917 ± 65.4	90.5 ± 5.5	64.2 ± 3.3	58.0 ± 3.8	1586 ± 471
10.0	103 ± 9.1	1030 ± 91.3	92.5 ± 7.2	64.7 ± 2.1	60.6 ± 4.0	1682 ± 209
11.0	102 ± 7.9	1128 ± 86.3	93.7 ± 6.6	65.7 ± 3.9	61.7 ± 5.4	1666 ± 112
12.0	102 ± 9.7	1197 ± 91.6	93.4 ± 4.8	65.8 ± 3.4	61.2 ± 40	1674 ± 220
7.0	100 ± 8.1	700 ± 56.4	78.6 ± 7.8	62.4 ± 3.9	49.7 ± 4.3	1541 ± 187
SEM ^1^	2.60	21.7	2.69	0.88	1.03	45.6

* Means differ in relation to the balanced feed (7 g/kg) by means of Dunnet test (*p* < 0.05). ^1^ Standard error of the mean.

**Table 11 animals-13-02659-t011:** Requirement for SID-Leucine for different proportions of population, based on maintenance requirement measured in Trial 1.

Scales	SID-Leu_m_ ^1^	50% of Population (σ = 0) ^2^	84% of Population (σ = 1) ^3^	97.7% of Population (σ = 2) ^4^
mg/kg	66	894	960	1025
mg/kg^0.75^	74	894	959	1025
mg/BP_m_^0.73^	395	977	1079	1180

^1^ SID-Leucine required for maintenance measured in Trial 1; ^2^ Mean requirement for the average individual in the population; ^3^ SID-Leucine intake to meet requirement of 84% of population; ^4^ SID-Leucine intake to meet requirement of 97.7% of population.

## Data Availability

No new data were created or analyzed in this study. Data sharing is not applicable to this article.

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
