# Peer review of "The Response to Dietary Leucine in Laying Lens"

_animals, 2023, doi:10.3390/ani13162659_

Round 1

Reviewer 1 Report

This is a very well written paper by an illustrious group of researchers and is novel covering leucine requirements in layers.  Well done on a beautifully written and very clear manuscript.  It was a real pleasure reading and reviewing this!

I only have a few minor comments regarding the document and, will do so by line number. 

Line 33: I think SID-Leu should read SID-Leui to define the acronym in the equation in line 34.

Line 74 to 76: Discusses digestible Leu as dLeu and the rest of the paper, digestible Leu is described as SID-Leu.  It is not clear if dLeu = SID-Leu but maybe keep to the SID-Leu throughout?  Not a big deal, however if you do not want to change this.

Line 106: Table 1 under ingredients has figures in parentheses that are not defined.  For corn, as an example, the value is (78.8 g/kg) and, whilst this is undoubtedly crude protein, it needs to be defined.  Also, the amino acids have their purity in percent and, that should be defined.  Again, it is unclear if the digestible amino acids such as Lysine dig. are SID values or not.  Maybe better to be consistent?

Line 111: Remove the statement following superscript "2" - There are no analysed amino acids in Table 1.

Line 190 & 191: Table 3. The ingredient crude proteins in parentheses now in percent and not consistent with Table 1 and again, need to be defined.  I suggest they get changed to g/kg.  Also, the added amino acids with the percent active are not defined.  Once again, maybe the SID term should be used rather than "dig."?

Line 195 & 196:  The superscript "2" needs to appear in the table at the "Calculated composition (g/kg)" position.

Line 198:  Table 4.  The superscrit "1" should appear next to the title of "Leucine (g/kg)" consistent with these in Tables, 5, 6 and 8.

Lines 244, 247, 250 and 253:  Tables 5, 6, 7 and 8- For consistency, the first column of these tables should have Leu levels as g/kg rather than %.

Line 278: Table 10 is correct with the first column as g/kg.  However, it should read "Leu levels (g/kg)" and not only "Levels (g/kg)".

Author Response

REVIEWER 1

This is a very well written paper by an illustrious group of researchers and is novel covering leucine requirements in layers.  Well done on a beautifully written and very clear manuscript.  It was a real pleasure reading and reviewing this!

Author's Answer: Dear Reviewer, we would like to thank you for your valuable contributions. We are sure that the quality of this manuscript was improved after this review. Please find below our responses; we tried our best to accommodate each one of the comments. We look forward to hearing back, and we hope the reviewer finds our modifications in line with the recommendations.

I only have a few minor comments regarding the document and, will do so by line number.

Line 33: I think SID-Leu should read SID-Leui to define the acronym in the equation in line 34.

Author's Answer: Done.

Line 74 to 76: Discusses digestible Leu as dLeu and the rest of the paper, digestible Leu is described as SID-Leu.  It is not clear if dLeu = SID-Leu but maybe keep to the SID-Leu throughout?  Not a big deal, however if you do not want to change this.

Author's Answer: We agree with the reviewers' suggestion and proper changes were performed in the text.

Line 106: Table 1 under ingredients has figures in parentheses that are not defined.  For corn, as an example, the value is (78.8 g/kg) and, whilst this is undoubtedly crude protein, it needs to be defined.  Also, the amino acids have their purity in percent and, that should be defined.  Again, it is unclear if the digestible amino acids such as Lysine dig. are SID values or not.  Maybe better to be consistent?

Author's Answer: We appreciate the referee’s suggestions. We included the information in the footnote.

Line 111: Remove the statement following superscript "2" - There are no analysed amino acids in Table 1.

Author's Answer: Done.

Line 190 & 191: Table 3. The ingredient crude proteins in parentheses now in percent and not consistent with Table 1 and again, need to be defined.  I suggest they get changed to g/kg.  Also, the added amino acids with the percent active are not defined.  Once again, maybe the SID term should be used rather than "dig."?

Author's Answer: Done. We changed the table and footnote to accommodate each observation.

Line 195 & 196:  The superscript "2" needs to appear in the table at the "Calculated composition (g/kg)" position.

Author's Answer: Done. Now it is the superscript “4”.

Line 198:  Table 4.  The superscrit "1" should appear next to the title of "Leucine (g/kg)" consistent with these in Tables, 5, 6 and 8.

Author's Answer: Done.

Lines 244, 247, 250 and 253:  Tables 5, 6, 7 and 8- For consistency, the first column of these tables should have Leu levels as g/kg rather than %.

Author's Answer: Done.

Line 278: Table 10 is correct with the first column as g/kg.  However, it should read "Leu levels (g/kg)" and not only "Levels (g/kg)".

Author's Answer: We changed to “Leu Content” to be consistent with previous tables.

Reviewer 2 Report

Manuscript is poorly written and sentence structure  is not clear. Abstract missing the experimental layout, methods used and most importantly the significant results of the study. Moreover, I don’t think 4 weeks data is reliable to fetch results from the current study. Furthermore, I am also unable to find the accurate P values in results tables. Overall results section is poorly presented.

Here are some suggestions to improve the manuscript

Line 14-16: sentence is not clear ‘It is notable that maintenance will be a smaller fraction of the daily nutrient requirement in growing animals than for laying hens, where the maintenance requirement is a significant proportion of the nutrient requirement’ please rewrite the sentence

Line 16-18: … egg production of layers or other? Please specify ‘For this reason, an accurate prediction of the maintenance  requirement is essential when partitioning the requirement between maintenance and egg production’

Line 28-23: nothing is clear ‘This study aimed to estimate the standardized ileal digestible leucine (SID-Leu) intake in laying hens for maintenance, and to describe the response in laying hens to SID-Leu thereby providing the information required to determine the optimum economic intake of SID-Leu for laying hens. Results showed that the amount of Leu required for egg production (11.6 mg/g egg) suggests an efficiency of utilization of dietary Leu of 0.79. Leu is unlikely to be limiting in laying hen feeds in which maize is the main cereal used, due to the high Leu content in this cereal’ please simplify the summary and make it more understandable.

Line 28: what were randomly distributed among six levels each replicated six times? Its not clear

Line 41: don’t start sentence with abbreviation ‘Leu increases the expression…’

Line 51: correct it ‘cited by Emmans and Fisher’

Line 51-53: correct it ‘t is notable that maintenance will be a smaller fraction of the daily nutrient requirement in growing animals than for laying hens, where the maintenance requirement is a significant proportion of the nutrient requirement [5]’

Line 66: this sentence is incomplete ‘The Reading Model [12] is such a model.’

Table 2: explain D1 to D6 in the foot note of the table

Similar in table 4

Line 237: don’t start the sentence with abbreviation ‘BW, SID-Leu intake, excretion a…

Line 237: nitrogen balance (NB) has already been abbreviated. Why you are abbreviating it again ?

Extensive English Editing is required 

Author Response

REVIEWER 2

Manuscript is poorly written and sentence structure  is not clear. Abstract missing the experimental layout, methods used and most importantly the significant results of the study. Moreover, I don’t think 4 weeks data is reliable to fetch results from the current study. Furthermore, I am also unable to find the accurate P values in results tables. Overall results section is poorly presented.

Author's Answers: Dear Reviewer, we after carefully discussing the reviewer's comments, we did some changes and also presented our points to support the statistical analysis approach used in this manuscript. The manuscript was completely reviewed in regard to improve the readability of the text.

 In regard to the 4 weeks data, there are at least six papers published in which the reading model has been used with laying hens. In each of these, the trial lasted ten weeks, with the final four weeks of data being used for analysis. The reason for this is that by the sixth week, the hens are in a relatively steady state on the feeds they have been fed, so the results of the final four weeks are better than those over the whole then-week period.

Please find below our responses; we tried our best to accommodate each one of the comments. We look forward to hearing back, and we hope the reviewer finds our modifications in line with the recommendations.

Here are some suggestions to improve the manuscript

Line 14-16: sentence is not clear ‘It is notable that maintenance will be a smaller fraction of the daily nutrient requirement in growing animals than for laying hens, where the maintenance requirement is a significant proportion of the nutrient requirement’ please rewrite the sentence

Author's Answers: The section was rewritten to accommodate the reviewer's suggestions.

Line 16-18: … egg production of layers or other? Please specify ‘For this reason, an accurate prediction of the maintenance  requirement is essential when partitioning the requirement between maintenance and egg production’

Author's Answers: The section was rewritten to accommodate the reviewer's suggestions.

Line 28-23: nothing is clear ‘This study aimed to estimate the standardized ileal digestible leucine (SID-Leu) intake in laying hens for maintenance, and to describe the response in laying hens to SID-Leu thereby providing the information required to determine the optimum economic intake of SID-Leu for laying hens. Results showed that the amount of Leu required for egg production (11.6 mg/g egg) suggests an efficiency of utilization of dietary Leu of 0.79. Leu is unlikely to be limiting in laying hen feeds in which maize is the main cereal used, due to the high Leu content in this cereal’ please simplify the summary and make it more understandable.

Author's Answers: The section was rewritten to accommodate the reviewer's suggestions.

Line 28: what were randomly distributed among six levels each replicated six times? Its not clear

Author's Answers: The section was rewritten to accommodate the reviewer's suggestions.

Line 41: don’t start sentence with abbreviation ‘Leu increases the expression…’

Author's Answers: Done.

Line 51: correct it ‘cited by Emmans and Fisher’

Author's Answers: Done.

Line 51-53: correct it ‘t is notable that maintenance will be a smaller fraction of the daily nutrient requirement in growing animals than for laying hens, where the maintenance requirement is a significant proportion of the nutrient requirement [5]’

Author's Answers: The text was rewritten to accommodate the reviewer's suggestion.

Line 66: this sentence is incomplete ‘The Reading Model [12] is such a model.’

Author's Answers: The text was rewritten to accommodate the reviewer's suggestion.

Table 2: explain D1 to D6 in the foot note of the table

Author's Answers: We changed the table so there is no need to include the footnote.

Similar in table 4

Author's Answers: We changed the table so there is no need to include the footnote.

Line 237: don’t start the sentence with abbreviation ‘BW, SID-Leu intake, excretion a…

Author's Answers: The text was rewritten to accommodate the reviewer's suggestion.

Line 237: nitrogen balance (NB) has already been abbreviated. Why you are abbreviating it again ?

Author's Answers: The text was rewritten to accommodate the reviewer's suggestion.

Reviewer 3 Report

The manuscript describes 2 nutrition experiments in which 1) the maintenance leucine (Leu) requirement was determined and 2) production responses to Leu (requirements) was determined for laying hens. The manuscript is generally well written and the experiments well conducted. I have but a few comments:

LINE                            COMMENT

76                                The very last part of the sentence (“2.”) should be deleted?

Table 1                         Footnote superscripts are missing for mineral and vitamin premixes 

Table 1,3 footnote 1      Please what the “kg” refers to (premix or diet). Please make sure the nutrients are per kilogram of complete, finished feed (not in the premix itself).

Table 1, footnote 1        Because the vitamin and trace mineral premixes were 2 different ingredients, we need 2 different footnotes, 1 for each of the premixes. 

Table 1, footnote 1        List the type of antioxidant (e.g., ethoxyquin)

Table 1, footnote 1        The way it is written, it is unclear if there is 500 mg/kg salinomycin or 500 mg/kg 12%salinomycing (we need to know how much active ingredient there was). 

Table 1, footnote 2        There are no values in parentheses in the table…

164+                            I have no idea what a counterproof treatment is… I have never heard the term. Maybe use a more common terminology?

Table 3                         Include a line for total amounts of ingredients

Table 3, footnote 1        Use correct spelling of iodine

209                              Typically, “egg mass” is used, not egg output

232-234                        I am not sure you need to include this paragraph…

238                              BPm needs to be defined

244+and Table 11         “Scale” is not the correct term. Please use ‘unit’ instead

Table 5+                       I am gussing you are going for 3 significant digits in the NB column. However, I suggest to round the 0.35% Leu data to the nearest whole number

Table 5+                       Include (in a footnote or in the caption) that values are means of n = ? observations

Figure 1                        The data points for 0 Leu intake are difficult to see. I suggest you have the Y-axis cross the X-axis at a Leu intake of  –100 

292+                            Many of the superscripts are not shown as superscripts (e.g., BW0.75). Please correct.

See comments above

Author Response

REVIEWER 3

The manuscript describes 2 nutrition experiments in which 1) the maintenance leucine (Leu) requirement was determined and 2) production responses to Leu (requirements) was determined for laying hens. The manuscript is generally well written and the experiments well conducted. I have but a few comments:

Author's answer: Dear Reviewer, we would like to thank you for your valuable contributions. We are sure that the quality of this manuscript was improved after this review. Please find below our responses; we tried our best to accommodate each one of the comments. We look forward to hearing back, and we hope the reviewer finds our modifications in line with the recommendations.

LINE  COMMENT

Line 76:  The very last part of the sentence (“2.”) should be deleted?

Author's answer: Done

Table 1 :   Footnote superscripts are missing for mineral and vitamin premixes

Author's answer: Done

Table 1,3:  footnote 1    Please what the “kg” refers to (premix or diet). Please make sure the nutrients are per kilogram of complete, finished feed (not in the premix itself).

Author's answer: Done

Table 1, footnote 1: Because the vitamin and trace mineral premixes were 2 different ingredients, we need 2 different footnotes, 1 for each of the premixes.

Author's answer: We changed the table and now vitamin and mineral premix are reported together, reflecting what was done in the trial.

Table 1, footnote 1:  List the type of antioxidant (e.g., ethoxyquin)

Author's answer: The antioxidant used was Butylated hydroxytoluene. The information was added in the footnote.

Table 1, footnote 1:  The way it is written, it is unclear if there is 500 mg/kg salinomycin or 500 mg/kg 12%salinomycing (we need to know how much active ingredient there was).

Author's answer: We rewrote the text to accommodate the reviewer's suggestion.

Table 1, footnote 2: There are no values in parentheses in the table…

Author's answer: The footnote was removed from table 1.

Line 164:  I have no idea what a counterproof treatment is… I have never heard the term. Maybe use a more common terminology?

Author's answer: We changed the term “counter-proof” to “balanced feed”.

Table 3:  Include a line for total amounts of ingredients

Author's answer: Done.

Table 3, footnote 1 : Use correct spelling of iodine

Author's answer: Done.

Line 209:   Typically, “egg mass” is used, not egg output

Author's answer: We believe that both terminologies are used in literature, and they represent the result of egg weight multiplied by egg production. Since most papers that are related to this work refer to this measure as egg output, we would like to keep the same terminology. Hopefully, the reviewer will agree with us.

Line: 232-234  -   I am not sure you need to include this paragraph…

Author's answer: For a better understanding of the reading model, we would like to keep it. Hopefully the reviewer will agree with us.

Line 238:  BPm needs to be defined

Author's answer: The term was defined at L-181.

Line 244 and Table 11:  “Scale” is not the correct term. Please use ‘unit’ instead

Author's answer: at line 179 we added a text defining “…scales, or units”. Hopefully, the reviewer will agree with us.

Table 5: I am gussing you are going for 3 significant digits in the NB column. However, I suggest to round the 0.35% Leu data to the nearest whole number

Author's answer: Done

Table 5:  Include (in a footnote or in the caption) that values are means of n = ? observations

Author's answer: Done

Figure 1:  The data points for 0 Leu intake are difficult to see. I suggest you have the Y-axis cross the X-axis at a Leu intake of  –100

Author's answer: Done

Line 292:  Many of the superscripts are not shown as superscripts (e.g., BW0.75). Please correct.

Author's answer: Done

Round 2

Reviewer 2 Report

Thanks for responding to my queries. I am still not satisfied with the contents, flow and English language of the manuscript.  

Language is too poor to understand the manuscript. Extensive English editing is required 

Author Response

Reviewer 2: Thanks for responding to my queries. I am still not satisfied with the contents, flow and English language of the manuscript. Language is too poor to understand the manuscript. Extensive English editing is required.

Author's answer: Respectfully, we desagree of your opnion. This paper was corrected by one of the authors (RMG) that has published over 250 papers in international journals and is a fluent English speaker and writer who has corrected the English in many dozens of manuscripts written by students and other academics and researchers.  Most of the 250 papers have been published in British Poultry Science, a journal known for its high quality of English.